# Transferable Normalization: Towards Improving Transferability of Deep Neural Networks

**Ximei Wang, Ying Jin, Mingsheng Long (✉),** *[*] **Jianmin Wang, and Michael I. Jordan**[♯]
School of Software, BNRist, Tsinghua University, China
Research Center for Big Data, Tsinghua University, China
National Engineering Laboratory for Big Data Software
[♯]University of California, Berkeley, Berkeley, USA
{wxm17,jiny18}@mails.tsinghua.edu.cn  {mingsheng,jimwang}@tsinghua.edu.cn
jordan@cs.berkeley.edu

## Abstract

Deep neural networks (DNNs) excel at learning representations when trained on large-scale datasets. Pre-trained DNNs also show strong transferability when fine-tuned to other labeled datasets. However, such transferability becomes weak when the target dataset is fully unlabeled as in Unsupervised Domain Adaptation (UDA). We envision that the loss of transferability mainly stems from the intrinsic limitation of the architecture design of DNNs. In this paper, we delve into the components of DNN architectures and propose Transferable Normalization (**TransNorm**) in place of existing normalization techniques. TransNorm is an end-to-end trainable layer to make DNNs more transferable across domains. As a general method, TransNorm can be easily applied to various deep neural networks and domain adaption methods, without introducing any extra hyper-parameters or learnable parameters. Empirical results justify that TransNorm not only improves classification accuracies but also accelerates convergence for mainstream DNN-based domain adaptation methods.

## 1   Introduction

Deep neural networks (DNNs) have dramatically advanced the state of the art in machine learning and excel at learning discriminative representations that are useful for a wide range of tasks [24, 3, 44]. However, in real-world scenarios where gaining sufficient labeled data through manual labeling is intolerably time-consuming and labor-exhausting, DNNs may be confronted with challenges when generalizing the pre-trained model to a different unlabeled dataset. This dilemma has motivated the research on domain adaptation [25], aiming to establish versatile algorithms that transfer knowledge from a different but related dataset by leveraging its readily-available labeled data.

Early domain adaptation methods strived to learn domain-invariant representations [25, 6] or instance importances [11, 5] to bridge the source and target domains. Since deep neural networks (DNNs) became successful, domain adaptation methods also leveraged them for representation learning. Disentangling explanatory factors of variations, DNNs are able to learn more transferable features [24, 3, 44, 47]. Recent works in deep domain adaptation embed discrepancy measures into deep architectures to align feature distributions across domains. Commonly, they minimize the distribution discrepancy between feature distributions [38, 17, 19, 20, 16] or adversarially learn transferable feature representations to fool a domain discriminator in a two-player game [36, 4, 37, 26, 18, 41, 46].

To date there have been extensive works that tackle domain adaptation by reducing the domain shift from the perspective of loss function designs. They are admittedly quite effective but constitute only

---

[*]Corresponding author: Mingsheng Long (mingsheng@tsinghua.edu.cn)

one side of the coin. Rare attention has been paid to the other side of the coin, i.e. **network design**, which is fundamental to deep neural networks and also an indispensable part of domain adaptation. Few previous works have delved into the transferability of the involved network backbone, of the internal components and most importantly, into the intrinsic limitation of these internal components.

To understand the impact of each internal component on the transferability of the whole deep networks, we delve into ResNet [9], one of the most elegant backbones commonly used in domain adaptation. In ResNet, convolutional layers contribute to feature transformation and abstraction, while pooling layers contribute to enhancing the translation invariance. However, the invariance they achieve cannot bridge the domain gap in domain adaptation since the domains are significantly different [13]. By contrast, Batch Normalization (BN) is crucial for speeding up training as well as enhancing model performance by smoothing loss landscapes [34]. It normalizes inputs to be zero-mean and univariance and then scales and shifts the normalized vectors with two trainable parameters to maintain expressiveness. Intuitively, BN shares the spirit of statistics (moments) matching [17] with domain adaptation.

However, when BN is applied to cross-domain scenarios, its design is not optimal for improving the transferability of DNNs. At each BN layer, the representations from the source and target domains are combined into a single batch and fed as the input of the BN layer. In this way, the sufficient statistics (including mean and standard deviation) of the BN layer will be calculated on this combined batch, without considering to which domain these representations belong. In fact, owing to the phenomenon known as *dataset bias* or *domain shift* [29], the distributions from source and target representations are significantly different, which implies a significant difference in their means and variances between domains. Hence, a brute-force sharing of the mean and variance across domains as in the BN layers may distort the feature distributions and thereby deteriorate the network transferability.

Further, the design that all channels in BN are equally important is unsuitable for domain adaptation. It is clear that some channels are easier to transfer than others since they constitute the sharing patterns of both domains. While those more transferable channels should be highlighted, it remains unclear how this can be implemented in an effective way, retaining the simplicity of BN while not changing other components of network architectures. Apparently, the inability to adapt representations according to channel transferability constitutes a major bottleneck in designing transferable architectures.

This paper delves into the components of DNN architectures and reveals that BN is the constraint of network transferability. We thus propose Transferable Normalization (**TransNorm**) in place of existing normalization techniques. TransNorm is an end-to-end trainable layer to make DNNs more transferable across domains. As a simple and general method, TransNorm can be easily plugged into various deep neural networks and domain adaption methods, free from changing the other network modules and from introducing any extra hyper-parameters or learnable parameters. Empirical results on five standard domain adaptation datasets justify that TransNorm not only improves classification accuracies but also accelerates convergence for state of the art domain adaptation methods on DNNs.

## 2   Related Work

**Domain Adaptation.**    Domain adaptation overcomes the shift across the source and target domains [25]. Common methods of domain adaptation fall into two classes: moment matching and adversarial training. Moment matching methods align feature distributions by minimizing the distribution discrepancy in the feature space, where DAN [17] and DDC [38] employ Maximum Mean Discrepancy [8] and JAN [20] uses Joint Maximum Mean Discrepancy. Adversarial training becomes very successful since the seminal work of Generative Adversarial Networks (GAN) [7], in which the generator and discriminator are improved through competing against each other in a two-player minimax game. Inspired by the idea of adversarial learning [7], DANN [4] introduces the domain discriminator which distinguishes the source and the target, while the feature extractor is trained to fool the discriminator. Since DANN was proposed, there were many attempts to further improve it. Motivated by Conditional GAN [23], CDAN [18] trains deep networks in a conditional domain-adversarial paradigm. MADA [26] extends DANN to multiple discriminators and pursues multimodal distribution alignment. ADDA [37] adopts asymmetric feature extractors for each domain while MCD [32] makes two classifiers consistent across domains. MDD [46] is the latest state of the art approach instantiating a new domain adaptation margin theory based on the hypothesis-induced discrepancy.

**Normalization Techniques.**    Normalization techniques are widely embedded in DNN architectures. Among them, Batch Normalization (BN) [12] has the strong ability to accelerate and stabilize training by controlling the first and second moments of inputs. It transforms inputs of every layer to have

zero-mean and univariance and then scales and shifts them with two trainable parameters to uncover the identity transform. Such a simple mechanism has been applied to most deep network architectures. At first, BN was believed to stabilize training by reducing the Internal Covariate Shift (ICS), while recent works proved that the efficacy of BN stems from smoother loss landscapes [34]. Meanwhile, BN has various variants, such as Layer Normalization [1] and Group Normalization [43].

There is no doubt that Batch Normalization is among the most successful innovations in deep neural networks, not only as a training method but also as a crucial component of the network backbone. However, in each BN layer, the representations from the source and target domains are combined into a single batch to feed as the input of the BN layer. In this way, BN shares mean and variance across two domains, which is clearly unreasonable due to *domain shift*. Researchers have realized this problem, and proposed AdaBN [15] and AutoDIAL [22]. AdaBN uses target statistics at inference to reduce the domain shift, but these statistics are excluded from the training procedure. AutoDIAL injects a linear combination of the source and target features to BN at the training stage, in which an extra parameter is involved in each BN layer as the trade-off between the source and target domains. Further, there has been no work on modeling the channel-wise transferability in deep neural networks.

## 3 Approach

This paper aims at improving transferability of deep neural networks in the context of Unsupervised Domain Adaptation (**UDA**). With a labeled source domain $\mathcal{D}_s = \{(x_{s,i}, l_{s,i})\}_{i=1}^{n_s}$ and an unlabeled target domain $\mathcal{D}_t = \{x_{t,i}\}_{i=1}^{n_t}$ where $x_i$ is an example and $l_i$ is the associated label, UDA usually trains a deep network that generalizes well to the unlabeled target data. The discussion in Section 1 elaborates the need of developing a Transferable Normalization (TransNorm) layer for the network backbone to enable domain adaptation, taking advantage of the *moment matching* mechanism in Batch Normalization (BN). The goal of TransNorm is to improve the transferability of deep networks with simple and efficient network layers easily pluggable into the network backbones of existing domain adaptation methods, by simply replacing their internal BN layers with our TransNorm layers. To smooth the presentation of TransNorm, we will first describe the preliminary of Batch Normalization.

### 3.1 Batch Normalization (BN)

Motivated by the wide practice that network training converges faster if its inputs are whitened [14, 42], Batch Normalization (BN) [12] was designed to transform the features of each layer to zero-mean and univariance. In order to reduce computation cost, it is implemented with two necessary simplifications: normalize *each scalar feature* (channel) independently in *each mini-batch*. First, it computes two basic statistics: **mean** $\mu^{(j)}$ and **variance** $\sigma^{2(j)}$ for each channel $j$ over a mini-batch of training data:

$$\mu^{(j)} = \frac{1}{m} \sum_{i=1}^{m} x_i^{(j)}, \quad \sigma^{2(j)} = \frac{1}{m} \sum_{i=1}^{m} (x_i^{(j)} - \mu^{(j)})^2, \tag{1}$$

where $m$ is the batch size. Then the statistics are used to normalize the input of each layer to have zero-mean and univariance. After that, a pair of learnable parameters **gamma** $\gamma^{(j)}$ and **beta** $\beta^{(j)}$ are learned for each channel $j$ to *scale and shift* the normalized value to uncover the identity transform:

$$\widehat{x}^{(j)} = \frac{x^{(j)} - \mu^{(j)}}{\sqrt{\sigma^{2(j)} + \epsilon}}, \quad y^{(j)} = \gamma^{(j)} \widehat{x}^{(j)} + \beta^{(j)}, \tag{2}$$

where $\epsilon$ is a constant added to the mini-batch variance for numerical stability. In this way, the Batch Normalizing Transform can successfully accelerate and stabilize training. However, it is somewhat unreasonable to combine the inputs of the source and target domains into a single batch since they follow different distributions. Furthermore, all channels are weighted with equal importance in BN, which is unsuitable for domain adaptation since some channels are more transferable than the others.

### 3.2 Transferable Normalization (TransNorm)

Towards the aforementioned challenges when applying Batch Normalization in domain adaptation, we propose Transferable Normalization (TransNorm) to improve the transferability of deep neural networks. Following the logistics of basic statistics, learnable parameters and intrinsic mechanism, we delve into BN step by step and replace its negative components towards transferability.

**Domain Specific Mean and Variance.** In existing domain adaptation methods, the representations from the source and target domains are combined into a batch to feed as the input of each BN layer.

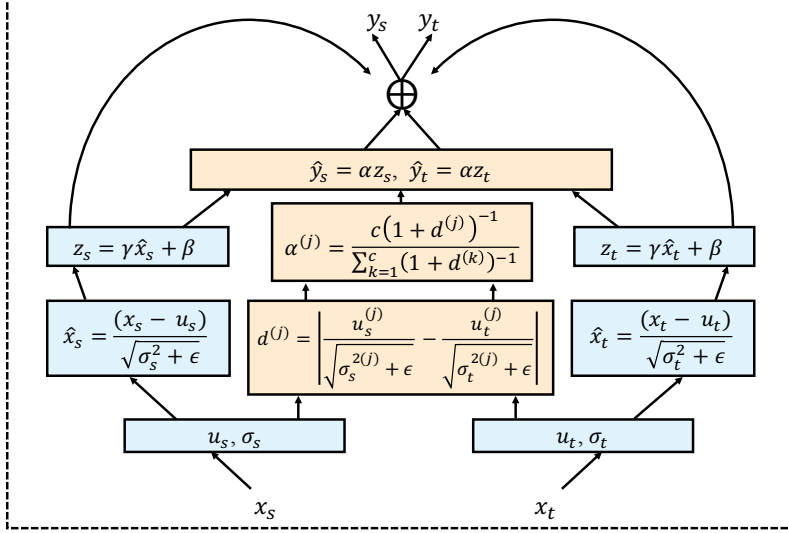

Figure 1: The architecture of Transferable Normalization (TransNorm).

However, due to the phenomenon known as *dataset bias* or *domain shift* [29], the representation distributions between source and target domains are quite different, leading to substantial difference between source statistics $u_s^{(j)}, \sigma_s^{2(j)}$ and target statistics $u_t^{(j)}, \sigma_t^{2(j)}$. Hence, a brute-force sharing of the mean and variance across domains will deteriorate transferability, making BN, the only component for statistics matching that potentially boosts transferability, fail to reduce the domain shift.

We mitigate this limitation of BN by normalizing the representations from source and target domains *separately* within each normalization layer. That is, the *domain specific* **mean** and **variance** of each channel are computed separately at first, and then in each layer we normalize the inputs independently for each domain as follows (lets focus on a particular activation $x^{(j)}$ and omit $j$ for clarity):

$$\widehat{x}_s = \frac{x_s - u_s}{\sqrt{\sigma_s^2 + \epsilon}}, \quad \widehat{x}_t = \frac{x_t - u_t}{\sqrt{\sigma_t^2 + \epsilon}}, \tag{3}$$

where $\epsilon$ is a constant added to the mini-batch variance for numerical stability. Appearing similar to AdaBN [15] that also separately calculates mean and variance of source and target domains, however, AdaBN only calculates statistics of the source domain at training, and then modulates that of target domain at inference. Admittedly, it can partially mitigate the limitation of BN, but the statistics of the target domain with important discriminative information remain unexploited at training. By contrast, both the statistics of source and target domains are applied in TransNorm at both training and inference, making it successfully capture the sufficient statistics of both domains.

**Domain Sharing Gamma and Beta.** With domain specific normalization, the inputs of TransNorm layer for each channel are normalized to have zero-mean and univariance. Note that simply normalizing each channel of a layer for different domains may lead to significant loss of expressiveness. Therefore, it is important to make sure that the transformation inserted in the neural network uncovers the identity transform which is shown to be indispensable in BN [12]. Akin to BN, we reuse the pair of learnable parameters **gamma** ($\gamma$) and **beta** ($\beta$) to *scale and shift* the normalized values as follows:

$$z_s = \gamma\widehat{x}_s + \beta, \quad z_t = \gamma\widehat{x}_t + \beta, \tag{4}$$

As the inputs of both domains are normalized to have zero-mean and univariance, the domain shift in the low-order statistics is partially reduced. Thus it becomes more reasonable to share parameters **gamma** ($\gamma$) and **beta** ($\beta$) across domains. This appears similar to AutoDIAL [22] which also shares these two parameters across domains. However, AutoDIAL requires extra learnable parameters to align the source and target domains while TransNorm requires no additional parameters.

**Domain Adaptive Alpha.** Both *domain specific mean and variance* and *domain sharing gamma and beta* process all channels uniformly. But it is intuitive that different channels capture different patterns. While some patterns are sharable across domains, others are not. Hence, each channel may embody different transferability across domains. A natural question arises: Can we quantify the

---

**Algorithm 1** Transferable Normalization (TransNorm)

---

**Input:** Values of $x$ in a mini-batch from source domain $\mathcal{D}_s = \{x_{s,i}\}_{i=1}^m$ and target domain
$\mathcal{D}_t = \{x_{t,i}\}_{i=1}^m$, the size $m$ of the mini-batch and the number $c$ of channels in each layer.
Parameters shared across domains to be learned: $\gamma$, $\beta$.
**Output:** $\{y_{s,i} = \text{TransNorm}_{\gamma,\beta}(x_{s,i})\}$, $\quad \{y_{t,i} = \text{TransNorm}_{\gamma,\beta}(x_{t,i})\}$.

$\mu_s \leftarrow \frac{1}{m}\sum_{i=1}^m x_{s,i}, \quad \mu_t \leftarrow \frac{1}{m}\sum_{i=1}^m x_{t,i}$  // mini-batch means of source and target domains

$\sigma_s^2 \leftarrow \frac{1}{m}\sum_{i=1}^m (x_{s,i} - \mu_s)^2, \quad \sigma_t^2 \leftarrow \frac{1}{m}\sum_{i=1}^m (x_{t,i} - \mu_t)^2$  // mini-batch variances for domains

$d^{(j)} \leftarrow \left| \frac{\mu_s^{(j)}}{\sqrt{\sigma_s^{2(j)}+\epsilon}} - \frac{\mu_t^{(j)}}{\sqrt{\sigma_t^{2(j)}+\epsilon}} \right|, \quad j = 1, 2, ..., c$  // statistics discrepancy for each channel

$\alpha^{(j)} \leftarrow \frac{c(1+d^{(j)})^{-1}}{\sum_{k=1}^c (1+d^{(k)})^{-1}}, \quad j = 1, 2, ..., c$  // transferability value for each channel

$\widehat{x}_s \leftarrow \frac{x_s - u_s}{\sqrt{\sigma_s^2+\epsilon}}, \quad \widehat{x}_t \leftarrow \frac{x_t - u_t}{\sqrt{\sigma_t^2+\epsilon}}$  // normalize the source and target domains separately

$y_{s,i} \leftarrow (1+\alpha)(\gamma\widehat{x}_{s,i}+\beta) \equiv \text{TransNorm}_{\gamma,\beta}(x_{s,i})$  // scale, shift and adapt in source domain

$y_{t,i} \leftarrow (1+\alpha)(\gamma\widehat{x}_{t,i}+\beta) \equiv \text{TransNorm}_{\gamma,\beta}(x_{t,i})$  // scale, shift and adapt in target domain

---

transferability of different channels and adapt them differently across domains? This paper gives a positive answer. We reuse the domain-specific statistics of means and variances *before normalization* to select those more transferable channels. These statistics, with channel-wise distribution information, enable us to quantify the domain distance $d^{(j)}$ for each channel $j$ in a parameter-free paradigm:

$$d^{(j)} = \left| \frac{\mu_s^{(j)}}{\sqrt{\sigma_s^{2(j)} + \epsilon}} - \frac{\mu_t^{(j)}}{\sqrt{\sigma_t^{2(j)} + \epsilon}} \right|, \quad j = 1, 2, ..., c, \tag{5}$$

where $c$ denotes the number of channels in the layer that TransNorm applies to. Reusing the previously calculated statistics, this calculation requires a negligible computational cost.

While the channel-wise distance $d^{(j)}$ naturally quantifies the inverse of transferability, we need to change it into a probabilistic weighting scheme. Motivated by the well-established t-SNE [21], we use *Student t-distribution* [35] with one degree of freedom to convert the computed distances to probabilities. This distribution, with *heavier tails* than alternatives such as Gaussian distribution, is suitable for highlighting transferable channels as well as avoiding overly penalizing the others. We compute the distance-based probability **alpha** ($\alpha$) to *adapt* each channel according to its transferability:

$$\alpha^{(j)} = \frac{c(1 + d^{(j)})^{-1}}{\sum_{k=1}^c (1 + d^{(k)})^{-1}}, \quad j = 1, 2, ..., c, \tag{6}$$

where $c$ is the channel size. In this way, the statistics computed before normalization are reused to quantify channel transferability, and finally, the channels with higher transferability will be assigned with higher importance for domain adaptation, enabling better learning of transferable patterns. To further avoid overly penalizing the informative channels, we use a residual connection to combine the normalized values in Eq. (4) with the transferability-weighted one to form the final output of the TransNorm layer:

$$y_s = (1 + \alpha)z_s, \quad y_t = (1 + \alpha)z_t. \tag{7}$$

**TransNorm Layer.** The overall process of TransNorm is summarized in Algorithm 1. Integrating the above explanation, TransNorm is designed to be an *end-to-end trainable* layer to improve the transferability of deep neural networks. As shown in Figure 1, TransNorm calculates statistics of inputs from the source and target domains separately while the channel transferability is computed *in parallel*. After normalized, features go through channel adaptive mechanisms to re-weight channels according to their transferability. It is quite crucial that the channel transferability strength shall be evaluated with statistics of *layer input* and assigned to the *normalized features*. In addition, the residual mechanism [9] is also included in TransNorm to make it more robust to subtle channels.

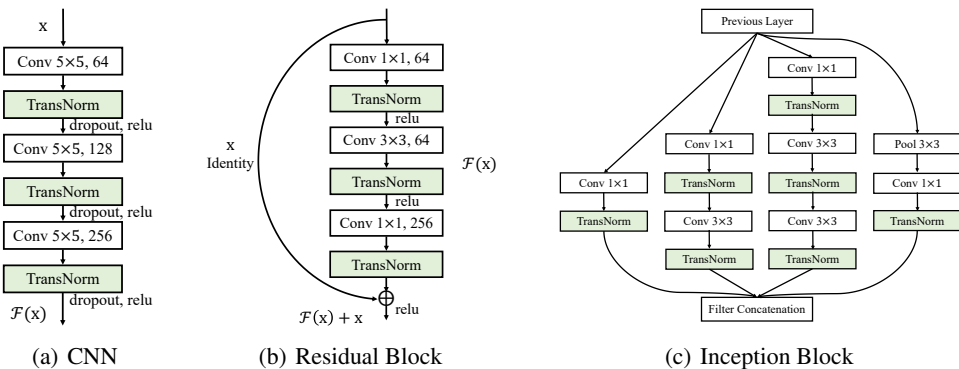

|  (a) CNN | (b) Residual Block | (c) Inception Block |

Figure 2: Improving transferability of mainstream deep backbones by replacing BN with TransNorm.

### 3.3 Improving Transferability of Deep Neural Networks

As a general and lightweight normalization layer, TransNorm can be easily plugged into common network backbones in place of the standard batch normalization, whenever they are pre-trained or *trained from scratch*. With stronger network backbones, most domain adaptation methods can be improved significantly without changing their loss functions. As shown in Figure 2, we can apply TransNorm to different kinds of deep neural networks, from vanilla CNN to ResNet to Inception Net, without any other modifications to the network architecture or introducing any additional parameters.

## 4 Experiments

We evaluate TransNorm with two series of experiments: **(a)** Basic experiments of applying TransNorm to the ResNet-50 backbone for seminal domain adaptation methods DANN [4] and CDAN [18] on three standard datasets; **(b)** Generalize to a wider variety of domain adaptation methods, more kinds of network backbones, and more challenging datasets. Experimental results indicate that our TransNorm can unanimously improve the transferability of the state of the art domain adaptation methods over five standard datasets. The code of TransNorm is available at `http://github.com/thuml/TransNorm`.

### 4.1 Setup

We use five domain adaptation datasets: **Office-31** [31] with 31 categories and 4,652 images collected from three domains: *Amazon* (**A**), *DSLR* (**D**) and *Webcam* (**W**); **ImageCLEF-DA** with 12 classes shared by three public datasets (domains): *Caltech-256* (**C**), *ImageNet ILSVRC 2012* (**I**), and *Pascal VOC 2012* (**P**); **Office-Home** [39] with 65 classes and 15,500 image from four significantly different domains: Artistic images (**Ar**), Clip Art (**Cl**), Product images (**Pr**) and Real-World images (**Rw**); **Digits** dataset with images from MNIST (**M**) and Street View House Numbers (SVHN, **S**); and **VisDA-2017** [27], a simulation-to-real dataset involving over 280K images from 12 categories. Our methods were implemented based on **PyTorch**, employing ResNet-50 [9] models pre-trained on the ImageNet dataset [30]. We follow the standard protocols for unsupervised domain adaptation[4, 20]. We conduct Deep Embedded Validation (**DEV**) [45] to select the hyper-parameters for all methods.

Table 1: Accuracy (%) on Office-31 for unsupervised domain adaption (ResNet-50).

| Method | A→W | D→W | W→D | A→D | D→A | W→A | Avg |
|---|---|---|---|---|---|---|---|
| ResNet-50 [9] | 68.4 ± 0.2 | 96.7 ± 0.1 | 99.3 ± 0.1 | 68.9 ± 0.2 | 62.5 ± 0.3 | 60.7 ± 0.3 | 76.1 |
| DAN [17] | 80.5 ± 0.4 | 97.1 ± 0.2 | 99.6 ± 0.1 | 78.6 ± 0.2 | 63.6 ± 0.3 | 62.8 ± 0.2 | 80.4 |
| RTN [19] | 84.5 ± 0.2 | 96.8 ± 0.1 | 99.4 ± 0.1 | 77.5 ± 0.3 | 66.2 ± 0.2 | 64.8 ± 0.3 | 81.6 |
| ADDA [37] | 86.2 ± 0.5 | 96.2 ± 0.3 | 98.4 ± 0.3 | 77.8 ± 0.3 | 69.5 ± 0.4 | 68.9 ± 0.5 | 82.9 |
| JAN [20] | 85.4 ± 0.3 | 97.4 ± 0.2 | 99.8 ± 0.2 | 84.7 ± 0.3 | 68.6 ± 0.3 | 70.0 ± 0.4 | 84.3 |
| MADA [26] | 90.0 ± 0.1 | 97.4 ± 0.1 | 99.6 ± 0.1 | 87.8 ± 0.2 | 70.3 ± 0.3 | 66.4 ± 0.3 | 85.2 |
| SimNet [28] | 88.6 ± 0.5 | 98.2 ± 0.2 | 99.7 ± 0.2 | 85.3 ± 0.3 | 73.4 ± 0.8 | 71.6 ± 0.6 | 86.2 |
| GTA [33] | 89.5 ± 0.5 | 97.9 ± 0.3 | 99.8 ± 0.4 | 87.7 ± 0.5 | 72.8 ± 0.3 | 71.4 ± 0.4 | 86.5 |
| **DANN+BN** [4] | 82.0 ± 0.4 | 96.9 ± 0.2 | 99.1 ± 0.1 | 79.7 ± 0.4 | 68.2 ± 0.4 | 67.4 ± 0.5 | 82.2 |
| **DANN+TransNorm** | **91.8 ± 0.4** | **97.7 ± 0.2** | **100.0 ± 0.0** | **88.0 ± 0.2** | **68.2 ± 0.2** | **70.4 ± 0.3** | **86.0** |
| **CDAN+BN** [18] | 94.1 ± 0.1 | 98.6 ± 0.1 | **100.0 ± 0.0** | 92.9 ± 0.2 | 71.0 ± 0.3 | 69.3 ± 0.3 | 87.7 |
| **CDAN+TransNorm** | **95.7 ± 0.3** | **98.7 ± 0.2** | **100.0 ± 0.0** | **94.0 ± 0.2** | **73.4 ± 0.4** | **74.2 ± 0.3** | **89.3** |

Table 2: Accuracy (%) on Image-CLEF for unsupervised domain adaption (ResNet-50).

| Method | I→P | P→I | I→C | C→I | C→P | P→C | Avg |
|---|---|---|---|---|---|---|---|
| ResNet-50 [9] | 74.8 ± 0.3 | 83.9 ± 0.1 | 91.5 ± 0.3 | 78.0 ± 0.2 | 65.5 ± 0.3 | 91.2 ± 0.3 | 80.7 |
| DAN [17] | 74.5 ± 0.4 | 82.2 ± 0.2 | 92.8 ± 0.2 | 86.3 ± 0.4 | 69.2 ± 0.4 | 89.8 ± 0.4 | 82.5 |
| **DANN+BN** [4] | 75.0 ± 0.3 | 86.0 ± 0.3 | **96.2** ± 0.4 | 87.0 ± 0.5 | 74.3 ± 0.5 | 91.5 ± 0.6 | 85.0 |
| **DANN+TransNorm** | **78.2** ± 0.4 | **89.5** ± 0.2 | 95.5 ± 0.2 | **91.0** ± 0.2 | **76.0** ± 0.2 | **91.5** ± 0.3 | **87.0** |
| **CDAN+BN** [18] | 77.7 ± 0.3 | 90.7 ± 0.2 | **97.7** ± 0.3 | 91.3 ± 0.3 | 74.2 ± 0.2 | 94.3 ± 0.3 | 87.7 |
| **CDAN+TransNorm** | **78.3** ± 0.3 | **90.8** ± 0.2 | 96.7 ± 0.4 | **92.3** ± 0.2 | **78.0** ± 0.1 | **94.8** ± 0.3 | **88.5** |

Table 3: Accuracy (%) on Office-Home for unsupervised domain adaption (ResNet-50).

| Method (S:T) | Ar:Cl | Ar:Pr | Ar:Rw | Cl:Ar | Cl:Pr | Cl:Rw | Pr:Ar | Pr:Cl | Pr:Rw | Rw:Ar | Rw:Cl | Rw:Pr | Avg |
|---|---|---|---|---|---|---|---|---|---|---|---|---|---|
| ResNet-50 [9] | 34.9 | 50.0 | 58.0 | 37.4 | 41.9 | 46.2 | 38.5 | 31.2 | 60.4 | 53.9 | 41.2 | 59.9 | 46.1 |
| DAN [17] | 43.6 | 57.0 | 67.9 | 45.8 | 56.5 | 60.4 | 44.0 | 43.6 | 67.7 | 63.1 | 51.5 | 74.3 | 56.3 |
| JAN [20] | 45.9 | 61.2 | 68.9 | 50.4 | 59.7 | 61.0 | 45.8 | 43.4 | 70.3 | 63.9 | 52.4 | 76.8 | 58.3 |
| **DANN+BN** [4] | **45.6** | 59.3 | 70.1 | 47.0 | 58.5 | 60.9 | 46.1 | 43.7 | 68.5 | 63.2 | 51.8 | 76.8 | 57.6 |
| **DANN+TransNorm** | 43.5 | **60.9** | **72.1** | **51.0** | **61.5** | **62.5** | **49.6** | **46.8** | **70.4** | **63.7** | **52.2** | **77.9** | **59.3** |
| **CDAN+BN** [18] | **50.7** | 70.6 | 76.0 | 57.6 | 70.0 | 70.0 | 57.4 | 50.9 | 77.3 | 70.9 | 56.7 | 81.6 | 65.8 |
| **CDAN+TransNorm** | 50.2 | **71.4** | **77.4** | **59.3** | **72.7** | **73.1** | **61.0** | **53.1** | **79.5** | **71.9** | **59.0** | **82.9** | **67.6** |

## 4.2 Basic Results

**Office-31.** As Table 1 shows, TransNorm enables DANN to surpass methods with more complicated networks and loss designs. Meanwhile, TransNorm can further improve CDAN, a competitive method.

**ImageCLEF-DA.** Table 2 reports the results on ImageCLEF-DA. TransNorm strengthens domain adaptation methods in most tasks, though the improvement is smaller due to more similar domains.

**Office-Home.** Table 3 also demonstrates significantly improved the results of TransNorm on Office-Home. This dataset is much more complicated, showing the efficacy of TransNorm on hard problems.

Table 4: Accuracy (%) on more adaptation methods, network backbones, and datasets.

| Adaptation Method | | Network Backbone | | | | Dataset | |
|---|---|---|---|---|---|---|---|
| Method | Avg | Method (Inception) | Avg | Method (DTN) | S→M | Method | VisDA |
| DANN [4] | 82.2 | Inception-BN [12] | 75.5 | CyCADA [10] | 90.4 | GTA [33] | 69.5 |
| CDAN [18] | 87.7 | AdaBN [15] | 76.7 | MCD [32] | 94.2 | MCD [32] | 69.7 |
| **MDD+BN** [46] | 88.8 | **DANN+BN** [4] | 82.1 | **DANN+BN** [4] | 73.9 | **DANN+BN** [4] | 63.7 |
| | | **DANN+TransNorm** | 83.2 | **DANN+TransNorm** | 77.0 | **DANN+TransNorm** | 66.3 |
| **MDD+TransNorm** | 89.5 | **CDAN+BN** [18] | 84.9 | **CDAN+BN** [18] | 89.2 | **CDAN+BN** [18] | 70.0 |
| | | **CDAN+TransNorm** | 86.0 | **CDAN+TransNorm** | 94.7 | **CDAN+TransNorm** | 71.4 |

## 4.3 Generalized Results

**More Adaptation Methods.** TransNorm can be applied to more sophisticated domain adaptation methods. To show this, we further evaluate it on MDD [46], the latest state of the art method. Results on Office-31 dataset in Table 4 (left column) indicate that TransNorm can further improve MDD.

**More Network Backbones.** We evaluate another two network backbones: Inception-BN and a smaller CNN network to ensure whether TransNorm can work well when *trained from scratch*. As shown in Table 4 (middle column), DANN and CDAN are both improved apparently. Enhanced by TransNorm, CDAN outperforms all other complicated methods in the difficult digit task (**S→M**).

**More Challenging Datasets.** We apply TransNorm with domain adaptation methods on large-scale datasets, VisDA-2017. As shown in Table 4 (right column), TransNorm is consistently effective.

**More Normalization Methods.** We throughly compare with other normalization techniques respectively on DANN [4] and CDAN [18]. As shown in Table 5, TransNorm consistently outperforms the two related methods based either on DANN or CDAN. Further, since AdaBN does not exploit the discriminative statistics of the target domain at training, CDAN+AdaBN performs worse than CDAN.

Table 5: Accuracy comparison with BN, AdaBN and AutoDIAL (%) on Office-31 (ResNet-50).

| Method | DANN [4] | | | | CDAN [18] | | | |
|--------|----|-------|----------|-----------|----|-------|----------|-----------|
| Normalization | BN | AdaBN | AutoDIAL | **TransNorm** | BN | AdaBN | AutoDIAL | **TransNorm** |
| A→W | 82.0 | 82.4 | 84.8 | **91.8** | 94.1 | 88.8 | 92.3 | **95.7** |
| D→W | 96.9 | **97.7** | **97.7** | 97.7 | 98.6 | 98.6 | 98.6 | **98.7** |
| W→D | 99.1 | 99.8 | **100.0** | 100.0 | **100.0** | 100.0 | 100.0 | 100.0 |
| A→D | 79.7 | 81.0 | 85.7 | **88.0** | 92.9 | 92.7 | 93.0 | **94.0** |
| D→A | **68.2** | 67.2 | 63.9 | **68.2** | 71.5 | 70.8 | 71.5 | **73.4** |
| W→A | 67.4 | 68.2 | 68.7 | **70.4** | 69.3 | 70.0 | 72.2 | **74.2** |
| Avg | 82.2 | 82.7 | 83.5 | **86.0** | 87.7 | 86.8 | 87.9 | **89.3** |

## 4.4 Insight Analyses

**Ablation Study.** The **domain adaptive alpha** highlight more transferable channels by calculating channel-wise transferability $\alpha$ in two steps: *Calculating Distance* in Eq. (5) and *Generating Probability* in Eq. (6). Calculating the distance only between means $\mu$ cannot capture the variance $\sigma$. While Wasserstein distance [40] uses both $\mu$ and $\sigma$ as $\|\mu_s - \mu_t\|_2^2 + (\sigma_s^2 + \sigma_t^2 - 2\sigma_s\sigma_t)$, the relative impact of $\mu$ and $\sigma$ are not balanced. Our strategy calculates the distance between means normalized by variance $\mu/\sqrt{\sigma^2 + \epsilon}$, which is consistent with BatchNorm [12] and yields the best results, as shown in Table 6. Using the same strategy $\mu/\sqrt{\sigma^2 + \epsilon}$ of calculating distance, we further examine the design of generating probability. Generating the distance-based probability to quantify the transferability of each channel, Softmax's ***winner-takes-all*** strategy is not suitable, while Gaussian's tail is not as heavy as Student-*t*. Only Student-*t* distribution has ***heavier tails*** that highlight transferable channels while avoiding overly penalizing the others, supported by the results in Table 6.

Table 6: Ablation study (%) on Office-31 with CDAN+TransNorm (ResNet-50).

| Type | Weight | Distance Type | | | Probability Type | | |
|------|--------|----|---------------------|---------------------------|---------|----------|-----------|
| Ablation | $\alpha = 1$ | $\mu$ | Wasserstein Distance | $\mu/\sqrt{\sigma^2 + \epsilon}$ | Softmax | Gaussian | Student-*t* |
| A→W | 94.6 | 95.0 | 94.5 | **95.7** | 94.9 | 94.8 | **95.7** |
| D→W | 98.6 | **98.7** | **98.7** | **98.7** | 97.9 | 98.6 | **98.7** |
| W→D | **100.0** | 100.0 | 100.0 | 100.0 | 99.8 | 100.0 | 100.0 |
| A→D | 93.4 | 93.0 | 91.0 | **94.0** | 91.4 | 93.1 | **94.0** |
| D→A | 71.5 | 72.8 | 72.6 | **73.4** | 69.7 | 72.4 | **73.4** |
| W→A | 72.9 | 73.7 | 73.6 | **74.2** | 73.2 | 73.0 | **74.2** |
| Avg | 88.5 | 88.9 | 88.4 | **89.3** | 87.9 | 88.7 | **89.3** |

Further, to verify that domain specific mean and variance are the most reasonable design, we set $\alpha$ to 1 in TransNorm. As shown in Table 6, results of TransNorm ($\alpha = 1$) are better than that of AutoDIAL, concluding that *domain-specific* mean and variance are better than *domain-mixed* ones adopted by AutoDIAL. AutoDIAL aims at learning domain *mixing* parameter but its learnable parameter $\alpha$ converges to $\alpha \approx 1$ (implying **domain-specific**) when the network becomes deeper, as shown in Figure 3(b) (ResNet-50), consistent with Figure 3 (Inception-BN) of its original paper [22].

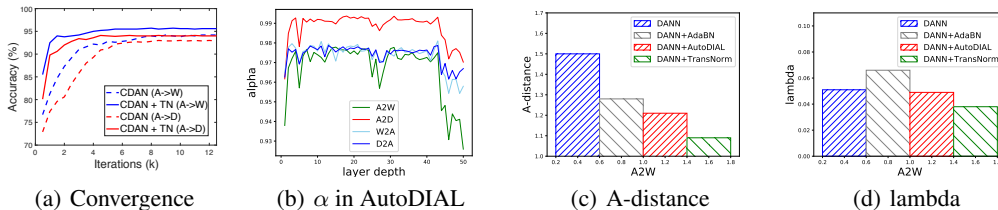

(a) Convergence     (b) $\alpha$ in AutoDIAL     (c) A-distance     (d) lambda

Figure 3: Visualization of convergence speed, $\alpha$ in AutoDIAL [22], A-distance and $\lambda$.

**Convergence Speed.** TransNorm can also accelerate convergence, as shown in Figure 3(a). Take CDAN on **A→W** as an example, it only needs $1,500$ iterations for CDAN with TransNorm to reach the accuracy of $95\%$, while at this point, the accuracy of CDAN with BN is still well below $90\%$.

**Theoretical Analysis.** Ben-David *et al.* [2] derived the learning bound of domain adaptation as $\mathcal{E}_{\mathcal{T}}(h) \leq \mathcal{E}_{\mathcal{S}}(h) + \frac{1}{2}d_{\mathcal{H}\Delta\mathcal{H}}(\mathcal{S}, \mathcal{T}) + \lambda$, which bounds the expected error $\mathcal{E}_{\mathcal{T}}(h)$ of a hypothesis $h$ on the target domain by three terms: **(a)** expected error of $h$ on the source domain, $\mathcal{E}_{\mathcal{S}}(h)$; **(b)** the A-distance $d_{\mathcal{H}\Delta\mathcal{H}}(\mathcal{S}, \mathcal{T}) = 2(1 - 2\epsilon)$, a measure of domain discrepancy, where $\epsilon$ is the error rate of a domain classifier trained to discriminate source domain and target domain; and **(c)** the error $\lambda$ of the ideal joint hypothesis $h^*$ on both source and target domains. By calculating A-distance and $\lambda$ on transfer task $\mathbf{A} \to \mathbf{W}$ with DANN, we found that our TransNorm can achieve **a lower A-distance** as well as **a lower** $\lambda$ (Figure 3), implying a lower generalization error by TransNorm.

| | Source Domain (Art) | Target Domain (Real World) |
|---|---|---|
| **Channel #05** $\alpha^{(05)} = 1.09$ | | |
| **Channel #39** $\alpha^{(39)} = 1.08$ | | |
| **Channel #32** $\alpha^{(32)} = 0.91$ | | |
| **Channel #48** $\alpha^{(48)} = 0.90$ | | |

In each group, **Left**: the original input image; **Middle**: the feature map of selected channel; **Right**: the combination of the input image and the selected channel.

Figure 4: Visualization of channels in the first convolutional layer of CDAN (TransNorm).

**Channel Visualization.** Though $\alpha$ in Eq. (6) is shared across domains, it is specific to each channel to highlight transferable channels. Here we calculate the transferability $\alpha$ for each channel after the training process converges. The channels with the largest and smallest transferability are shown in Figure 4. The channels with higher transferability capture more sharing patterns such as textures of objects while those with lower transferability focus on the background or varying patterns.

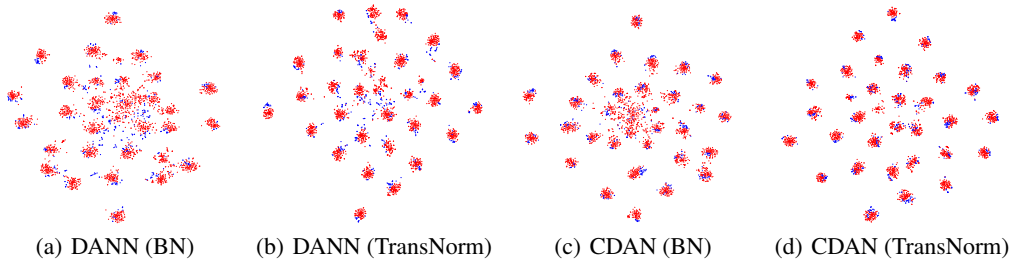

(a) DANN (BN)      (b) DANN (TransNorm)      (c) CDAN (BN)      (d) CDAN (TransNorm)

Figure 5: Visualization of features from the models with BN and TransNorm. (red: **A**; blue: **D**).

**Feature Visualization.** We use the t-SNE tool [21] to visualize the representations in the bottleneck layer of ResNet-50. We compare the models with BN and with TransNorm, based on DANN and CDAN on task $\mathbf{A} \to \mathbf{D}$ of Office-31. As shown in Figure 5, the source and target features in models with TransNorm are more indistinguishable, implying more transferable features learned by TransNorm. Meanwhile, class boundaries are sharper, proving that TransNorm learns more discriminative features.

## 5 Conclusion

We presented TransNorm, a simple, general and end-to-end trainable normalization layer towards improving the transferability of deep neural networks. It can be simply applied to a variety of network backbones by solely replacing batch normalization. Empirical studies show that TransNorm significantly boosts existing domain adaptation methods, yielding higher accuracy and faster convergence.

## Acknowledgments

This work was supported by the National Key R&D Program of China (2017YFC1502003) and the Natural Science Foundation of China (61772299 and 71690231).

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
