[Reviews · NeurIPS 2019]

Reviewer 1



Originality: TransNorm follows similar style from AdaBN and AutoDIAL, but shows higher performance. Clarity: the paper is well written. I especially like Figure 1, which clearly illustrate the proposed TransNorm. Quality: The proposed technique seems sound, and experimental settings/results are solid. Significance: The algorithms is easy to use (so as other normalization techniques). Though higher performance is shown in the paper, it is unknown how significant it is when adopting in other real scenarios compared with other normalization techniques.

Reviewer 2



This paper is different from most works focus on reducing the domain shift from perspective of loss functions, contributing to network design by developing a novel transferable normalization (TranNorm) layer. TranNorm is well motivated, separately normalizing source and target features in a minibatch and meanwhile weighting each channel in terms of transferability. It is clear different from and meanwhile significantly outperformes related methods, e.g., AdaBN [15] and AutoDIAL [21]. The TranNorm layer is simple and free of parameters, which can be conveniently plugged in mainstream networks. I think that this work will have a non-trivial impact: the proposed TranNorm can be used as backbone layer improving other state-of-the-art methods. The experiments are extensively evaluated both qualitatively and quantitively, demonstrating the effectiveness of the proposed TranNorm. The TranNorm layer is key contribution of this paper. This layer consists of separately normalizing the source and target features, followed by weighting with $\alpha$ with respect to transferability, which is empirically defined. I would like to see ablation analysis on $\alpha$: what will the performance be if one sets $\alpha=1$, and what will the performance be if the transferability is defined as softmax or Gaussian with tunable variance over discrepancy of statistics, i.e., $\mu/\sqrt{\sigma^2+\epsilon}$ (rather than only $\mu$). --------------------------------------------------------- One of my comments is that what the performance will be if the two probabilities build upon $\mu/\sqrt{\sigma^2+\epsilon}$, rather than only $\mu$. However, after reading the rebuttal, I still cannot see what kind of distances are used in the probabilities of softmax and Gaussian, from the second table and analysis therein. Note that I check the submitted code in trans_norm.py (lines 152 and 156), finding that the two probabilities build upon only $\mu$. I wish the authors make further clarification and perform corresponding experiments in the final version. I keep my original recommendation unchanged.

Reviewer 3



Strengths: + Proposes a new normalisation statistics-based method for DA. This line of attack against the domain-adaptation problem is rather under-studied compared to other approaches. + A good range of benchmarks. Evaluation on multiple base DA methods and network backbones. + Analysis Sec 4.4 is interesting. Weaknesses: 1. Weak novelty. Addressing domain-shift via domain specific moments is not new. It was done among others by Bilen & Vedaldi, 2017,”Universal representations: The missing link between faces, text, planktons, and cat breeds”. Although this paper may have made some better design decisions about exactly how to do it. 2. Justification & analysis: A normalisation-layer based algorithm is proposed, but without much theoretical analysis to justify the specific choices. EG: Why is is exactly: that gamma and beta should be domain-agnostic, but alpha should be domain specific. 3. Positioning wrt AutoDial, etc: The paper claims “parameter-free” as a strength compared to AutoDIAL, which has a domain-mixing parameter. However, this spin is a bit misleading. It removes one learnable parameter, but instead includes a somewhat complicated heuristic Eq 5-7 governing transferability. It’s not clear that removing a single parameters (which is learned in AutoDIAL) with a complicated heuristic function (which is hand-crafted here) is a clear win. 4. The evaluation is a good start with comparing several base DA methods with and without the proposed TransferNorm architecture. It would be stronger if the base DA methods were similarly evaluated with/without the architectural competitors such as AutoDial and AdaBN that are direct competitors to TN. 5. English is full of errors throughout. "Seldom previous works", etc. ------ Update ----- The authors response did a decent job of responding to the concerns. The paper could be reasonable to accept. I hope the authors can update the paper with the additional information from the response.

[Author Response · NeurIPS 2019]

We thank the reviewers for insightful and constructive comments. We have submitted **code** for full reproducibility.

**Common Questions:**

**Q1. Consistently compare with AdaBN and AutoDIAL in all the experiments.**

**A1.** Besides ablation result already given in Figure 5, we show complete results in the table below. **TransNorm (TN)**
significantly improves upon AdaBN and AutoDIAL. Without exploiting the discriminative statistics of the target domain
at training, CDAN+AdaBN performs worse than CDAN. We will analyze why TransNorm is better than AutoDIAL.

| Normalization | DANN [3] | DANN+AdaBN | DANN+AutoDIAL | **DANN+TN** | CDAN [17] | CDAN+AdaBN | CDAN+AutoDIAL | **CDAN+TN** |
|---|---|---|---|---|---|---|---|---|
| A→W | 82.0 | 82.4 | 84.8 | **91.8** | 94.1 | 88.8 | 92.3 | **95.7** |
| D→W | 96.9 | 97.7 | 97.7 | **97.7** | 98.6 | 98.6 | 98.6 | **98.7** |
| W→D | 99.1 | 99.8 | 100.0 | **100** | 100.0 | 100.0 | 100.0 | **100.0** |
| A→D | 79.7 | 81.0 | 85.7 | **88.0** | 92.9 | 92.7 | 93.0 | **94.0** |
| D→A | 68.2 | 67.2 | 63.9 | **68.2** | 71.5 | 70.8 | 71.5 | **73.4** |
| W→A | 67.4 | 68.2 | 68.7 | **70.4** | 69.3 | 70.0 | 72.2 | **74.2** |
| Avg | 82.2 | 82.7 | 83.5 | **86.0** | 87.7 | 86.8 | 87.9 | **89.3** |

**Q2. Why transferability $\alpha$ is designed in this way? Comparison with other types of transferability design.**

**A2.** The domain adaptive alpha aims to highlight more transferable channels by calculating channel-wise transferability
$\alpha$ in two steps: for each channel, **(a) Calculating Distance** in Eq. (5) and **(b) Generating Probability** in Eq. (6).

**(a)** Calculating the distance only between means $\mu$ cannot capture the variance $\sigma$. While Wasserstein distance uses
both $\mu$ and $\sigma$, the relative impact of $\mu$ and $\sigma$ are not well balanced. Our strategy calculates the distance between means
normalized by variance $\mu/\sqrt{\sigma^2 + \epsilon}$, which is consistent with BatchNorm [11] and yields the best results (table below).

**(b)** Generating the distance-based probability to quantify the transferability of each channel, Softmax's ***winner-takes-all***
strategy is not suitable, while Gaussian's tail is not as heavy as Student-t. Only Student-t distribution has ***heavier tails***
that highlight transferable channels while avoiding overly penalizing the others, supported by the results (table below).

| Type | Ablation | | Distance Type | | | Probability Type | | | Hand Keypoint Estimation | | |
|---|---|---|---|---|---|---|---|---|---|---|---|
| Abalation | $\alpha = 1$ | $\mu$ | Wasserstein Distance | $\mu/\sqrt{\sigma^2 + \epsilon}$ | Softmax | Gaussian | Student | Method | PCK@0.2 | PCK@0.05 |
| A→W | 94.6 | 95.0 | 94.5 | **95.7** | 94.9 | 94.8 | **95.7** | JAN [19] | 77.00 | 26.70 |
| D→W | 98.6 | 98.7 | 98.7 | **98.7** | 97.9 | 98.6 | **98.7** | DANN [3] | 80.10 | 29.61 |
| W→D | 100.0 | 100.0 | 100.0 | **100.0** | 99.8 | 100.0 | **100.0** | **DANN+TN** | 81.00 | 30.80 |
| A→D | 93.4 | 93.0 | 91.0 | **94.0** | 91.4 | 93.1 | **94.0** | MCD [30] | 80.15 | 30.10 |
| D→A | 71.5 | 72.8 | 72.6 | **73.4** | 69.7 | 72.4 | **73.4** | MCD+AutoDIAL | 80.38 | 30.51 |
| W→A | 72.9 | 73.7 | 73.6 | **74.2** | 73.2 | 73.0 | **74.2** | MCD+TN ($\alpha = 1$) | 80.80 | 31.10 |
| Avg | 88.5 | 88.9 | 88.4 | **89.3** | 87.9 | 88.7 | **89.3** | **MCD+TN** | **82.12** | **32.42** |

15
**Q3. Why domain specific mean and variance? What the performance will be if $\alpha$ is set to $1$.**

**A3.** Result of TransNorm ($\alpha = 1$) is better than that of AutoDIAL (above table), concluding that ***domain-specfic*** mean
and variance are better than ***domain-mixed*** ones by AutoDIAL. AutoDIAL learns *mixing* parameter which converges to
$\alpha \approx 1$ (implying domain-specific) in Figure 3 (Inception-BN) of its original paper and in the figure below (ResNet-50).

**R1/1. The performance of TransNorm when applying into other real scenarios.**

We apply TransNorm to a challenging regression task: **Hand Keypoint Estimation** for domain adaptation. The above
table shows transfer results from *CMU Panoptic Dataset* to *Rendered HandKeypoint Dataset* (depicted in figure below).
Note that, MCD+AutoDIAL performs worse than MCD+TN ($\alpha = 1$), validating that its alpha does not work generally.

**R4/1. Addressing domain-shift via domain specific moments is not new.**

Most existing works such as MMD [16] address domain-shift via *domain specific moments*, serving as the theoretical
foundation of our approach. We further provide an architecture perspective: A transferable normalization layer. It can
work with all deep domain adaptation methods. Further, as Reviewer #1 points that our *Domain Adaptive Alpha* is new.

**R4/2. Why is exactly that gamma and beta should be domain-agnostic, but alpha should be domain specific.**

As justified in BatchNorm [11] and ResNet [8], $\beta$ and $\gamma$ should be shared across domains to uncover the *identity map*.
We clarify that $\alpha$ is also shared across domains, although it is specific to each channel to highlight transferable channels.

**R4/3. Theoretical analysis to justify the specific design choices.**

Ben-David *et al.* gave the learning bound of domain adaptation as $\mathcal{E}_\mathcal{T}(h) \leq \mathcal{E}_\mathcal{S}(h) + \frac{1}{2}d_{\mathcal{H}\Delta\mathcal{H}}(\mathcal{S}, \mathcal{T}) + \lambda$. By calculating
A-Distance $d_{\mathcal{H}\Delta\mathcal{H}}(\mathcal{S}, \mathcal{T})$ and $\lambda$ on transfer task A → W, we found that our TransNorm can achieve **a lower A-distance**
and **a lower** $\lambda$ and thus guarantees a lower generalization error. We will add these theoretical analyses to the revision.



[Meta-Review · NeurIPS 2019]

The paper proposes a novel normalization strategies in the line of AdaBN and AutoDIAL paper. The method is simple, easy to use and shows better results than other normalization strategies. The paper is well written, the experimental evaluation is large and solid with the help of the complementary information provided in the rebuttal. Authors should precise better f distances are used in the probabilities of softmax and Gaussian